# Efficient Combination of Rematerialization and Offloading for Training DNNs

**Olivier Beaumont**         **Lionel Eyraud-Dubois**         **Alena Shilova**

Inria Bordeaux

{olivier.beaumont, lionel.eyraud-dubois, alena.shilova}@inria.fr

## Abstract

Rematerialization and offloading are two well known strategies to save memory during the training phase of deep neural networks, allowing data scientists to consider larger models, batch sizes or higher resolution data. Rematerialization trades memory for computation time, whereas Offloading trades memory for data movements. As these two resources are independent, it is appealing to consider the simultaneous combination of both strategies to save even more memory. We precisely model the costs and constraints corresponding to Deep Learning frameworks such as PyTorch or Tensorflow, we propose optimal algorithms to find a valid sequence of memory-constrained operations and finally, we evaluate the performance of proposed algorithms on realistic networks and computation platforms. Our experiments show that the possibility to offload can remove one third of the overhead of rematerialization, and that together they can reduce the memory used for activations by a factor 4 to 6, with an overhead below 20%.

## 1   Introduction

A general trend in machine learning, especially in NLP today, is to increase the size of models to improve accuracy. For example, NLP models like BERT [7] or GPT-3 [30] have billions of parameters and vision networks like EfficientNet [35], AmoebaNet [34] or SENET [15] routinely have a hundred million parameters, which raises difficult memory issues on architectures like GPUs. Moreover, it was observed [29] that a sufficiently large batch-size is required to obtain good and fast convergence. In this paper we focus on the strategies for reducing the memory usage of activations that can be implemented when considering a single computation node, typically a single GPU. These techniques can of course be combined with the different possible parallelizations of the training phase, like data parallelism [39, 27, 11], model parallelism [25, 37, 26], filter or kernel [9] or image parallelisms [8]. The combination with parallel strategies is however out of the scope of the present paper and our goal here is to model, analyze the complexity and understand how to benefit simultaneously from rematerialization and offloading.

Rematerialization [14, 5, 20, 3] is a technique that consists in deleting from memory some of the activations computed during the forward phase as soon as they are no longer needed for the forward phase. As these activations are required again during the backward phase, they are then recomputed from other activations kept in memory. The idea is therefore to trade additional computations for lower memory requirements. The objective, in the context of rematerialization, is to compute a sequence of valid operations (i.e. such that the inputs of any operation are actually in memory at the time of the operation) consisting of elementary forward, backward and delete operations, and which has a minimum execution time among all sequences that satisfy a given memory constraint. Offloading [32, 2, 23, 4] is a technique that has the same objective, *i.e.* to produce a sequence of operations satisfying a given memory constraint. However, in the case of offloading, each computation operation is executed exactly once. The idea here is therefore to trade transfers on the PCI bus between

the CPU and the GPU for lower memory requirements. The objective, in the context of offloading, is therefore to compute a sequence of valid operations consisting of elementary forward, backward, offload and prefetch operations, and which has a minimum execution time among all sequences that fulfill a given memory constraint.

While rematerialization and offloading share the common goal of limiting memory consumption, they rely on different techniques and achieve this result by consuming independent resources, using either more computation or more transfers between a GPU and a CPU. In this paper we solve the problem of finding the optimal sequence combining rematerialization and activation offloading, which has not been addressed in the literature to the best of our knowledge. We discuss related work in Section 2 and model and notations in Section 3. The algorithmic solution to combine optimally checkpointing and offloading is presented in Section 4 and experiments to assess the advantage of using both offloading and rematerialization, based on a large range of neural networks and architecture parameters, are presented in Section 5.

## 2 Related work

**Parallelism-based memory optimization**   A first approach to control memory consumption is to use parallelization. Two main sources of memory consumption in the training phase for deep neural networks are, on the one hand, the storage of the network weights and, on the other hand, the storage of activations. To limit the storage cost of activations (for a fixed batch size), data parallelism [39, 27, 11] is the method of choice. Each batch is divided into mini-batches which are in turn processed in parallel on different computation resources. In this scheme, however, weights are replicated on each resource and collective communications are required to synchronize weights between different resources. When models become large, these collective communications can hinder the scalability of the data parallel approach.

On the contrary, model parallelism [6, 17, 25, 37] is used to distribute weights over different available resources. Weights of a neural network are then distributed on different computation resources, and communications happen between consecutive layers assigned to different resources. Nevertheless, this approach also requires sophisticated strategies [26] to save fewer activations and models in memory. The data can also be split along other dimensions. In [8], large images are split into smaller images, the network is trained on these small images (augmented by a halo) and extra communications are needed in order to synchronize parameter updates. A similar strategy was proposed to use channel and filter parallelism in [9]. All these parallel strategies can be combined with the single node memory optimization strategies considered in the present paper.

**Rematerialization-based strategies**   Rematerialization techniques consist in keeping only a subset of activations in memory during the whole training phase, while the others are dynamically recomputed at runtime when they are needed for the backpropagation. This allows to explore a tradeoff between memory usage and computational cost. Rematerialization was first considered in the context of Automatic Differentiation (AD) [12] under the name "checkpointing". In the context of AD, computational networks consist in very long and homogeneous chains, in which the forward activation corresponding to the $i-$th stage of the chain has to be kept into memory until the $i-$th backward stage. Rematerialization algorithms consist in determining in advance which forward checkpoints should be kept into memory and which ones should be recomputed from stored checkpoints when performing the backward phase. In the case of homogeneous chains closed form formulas providing the exact position of checkpoints were proposed in [13].

The use of rematerialization strategies was advocated for DNN in several papers [14, 5, 20]. An implementation of a simple periodic strategy [5] was provided in PyTorch [28][1], but it is only optimal for the restricted case of homogeneous chains and when activation discards are done only before the loss. However, DNN models are in general more complicated. More recently, this strategy has been extended to general heterogenous networks represented as sequential networks in [3][2].

The case of non sequential networks was also addressed in the literature [19, 21, 18, 22, 10], mostly under the simplifying assumption that discarding computed activations is only possible before the computation of the loss function [22, 10]. In [19] the authors proposed an algorithm

---

[1]`https://pytorch.org/docs/stable/_modules/torch/utils/checkpoint.html`
[2]`https://gitlab.inria.fr/hiepacs/rotor`

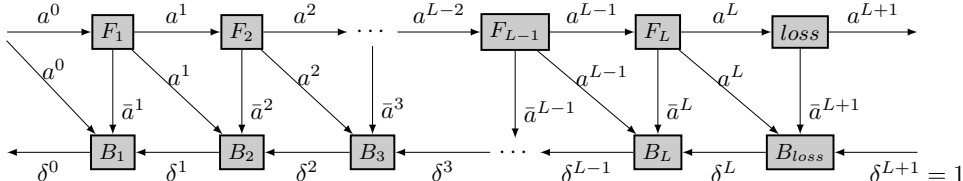

Figure 1: Data dependencies induced the training phase of Sequential Deep Neural Networks.

dealing with dynamic data flows, though based on an heuristical approach. The authors of [18] considered minimization of the runtime with the use of Integer Linear Programming. Even though this approach is suitable for arbitrary computational graphs with heterogeneous costs, solving this ILP is computationally very expensive even for intermediate depth networks.

**Offloading-based strategies** Offloading [32, 2] is an alternative approach which consists in offloading some of the forward activations from the memory of the GPU to the memory of the CPU, which is expected to be much larger. In [32], the authors proposed a simple mechanism of memory virtualization, that introduces unnecessary synchronizations between data transfers and computations of later forward activations. This technique was improved in [2]. In both papers, the algorithmic strategies to decide which activations to offload into the main memory are relatively straightforward, and basically consist in sending to the CPU either all activations or only those of convolutional layers. In [4], the complexity of the offloading problem was analyzed and several NP-completeness results were provided, depending whether activations can be offloaded partially or not.

In order to reduce the overhead incurred by the communications, the use of compression was advocated in [33]. Memory virtualization was further considered in [24, 23, 16, 38]. Finally, the speed-centric rematerialization from [5] enhanced with memory-centric rematerialization (discards activations of every segment all the time) was combined with the simple offloading approach from [32] in [36]. In addition, the authors in [31] also used rematerialization of [5] with a possibility of further offloading saved checkpoints to the CPU if rematerialization only is not enough to perform training under memory constraint.

## 3 Model and notations

### 3.1 Forward, backward and offloading elementary operations

In what follows, our goal is to process the chain described by Figure 1 that corresponds to the processing of a mini-batch. This chain depicted by Figure 1 is defined by the set of operations $F_\ell$ and $B_\ell$. The default forward operations $F_\ell$ requires only the activation produced by the previous layer $F_{\ell-1}$ (or the input mini-batch in the case of $F_1$), whereas the default backward operation $B_\ell$ require the preceding gradient $\delta^\ell$, the activation $a^{\ell-1}$ and the intermediate generated data $\bar{a}^\ell$ necessary for the backward step. The distinction between $a^\ell$ and $\bar{a}^\ell$ is crucial in practice, since it allows to consider computation graphs which have a sequential structure, but are not purely sequential. In this setting, it is indeed possible for $F_\ell$ to represent a complex operation (any Direct Acyclic Graph of layers), and thus $\bar{a}^\ell$ contains all the intermediate activations produced by $F_\ell$ and necessary to compute $B_\ell$, whereas $a^\ell$ only contains the output activation of $F_\ell$, that will be used by $F_{\ell+1}$. In general, $a^\ell$ can therefore be much smaller than $\bar{a}^\ell$.

In order to manipulate different types of activations, we define in Table 1 a set of generic forward and backward operations, that can be directly translated in automatic differentiation packages such as `tf.GradientTape` for Tensorflow or `torch.autograd` for PyTorch. Operations are defined both by the type of operands they use and produce ($a^\ell$ or $\bar{a}^\ell$) and by the treatment they make of the input data after processing (deleted or kept in memory). Each operation is associated to a processing time and a memory overhead that takes place during the computation. Both the processing time and the memory overhead only depend on whether the operation is a forward or a backward task.

In addition to the operations described in Table 1, we also consider two operations $O_\ell$ and $P_\ell$ that correspond respectively to offloading to and prefetching from CPU memory. $O_\ell$ and $P_\ell$ can operate

| | Operation | Input | Output | Time | Memory overhead |
|---|---|---|---|---|---|
| $F_{all}^\ell$ | Forward and save all | $\{a^{\ell-1}\}$ $\{\bar{a}^{\ell-1}\}$ | $\{a^{\ell-1},\ \bar{a}^\ell\}$ $\{\bar{a}^{\ell-1},\ \bar{a}^\ell\}$ | $u_{F_\ell}$ | $o_\ell^f$ |
| $F_{ck}^\ell$ | Forward and materialize input | $\{a^{\ell-1}\}$ $\{\bar{a}^{\ell-1}\}$ | $\{a^{\ell-1},\ a^\ell\}$ $\{\bar{a}^{\ell-1},\ a^\ell\}$ | $u_{F_\ell}$ | $o_\ell^f$ |
| $F_{\varnothing}^\ell$ | Forward without saving | $\{a^{\ell-1}\}$ $\{\bar{a}^{\ell-1}\}$ | $\{a^\ell\}$ $\{a^\ell\}$ | $u_{F_\ell}$ | $o_\ell^f$ |
| $B^l$ | Backward step | $\{\delta^\ell,\ \bar{a}^\ell,\ a^{\ell-1}\}$ $\{\delta^\ell,\ \bar{a}^\ell,\ \bar{a}^{\ell-1}\}$ | $\{\delta^{\ell-1}\}$ $\{\delta^{\ell-1},\ \bar{a}^{\ell-1}\}$ | $u_{B_\ell}$ | $o_\ell^b$ |

Table 1: Generic operations available in DL frameworks.

both on $a^{\ell-1}$ and $\bar{a}^{\ell-1}$, depending on which type of activation has been produced. Thanks to CUDA streams, it is possible to asynchronously offload and prefetch activations from and to the CPU memory, while independent computations are performed.

### 3.2 Assumptions and limitations

In what follows, our goal is to find a sequence of $F_{all}^\ell$, $B_\ell$, $F_{ck}^\ell$, $F_{\varnothing}^\ell$, $O_\ell$ and $P_\ell$ operations to compute $\delta^0$ from $a^0$. Given a bandwidth $\beta$ between the GPU and the CPU, we can compute the peak memory usage and the total duration of a particular sequence. Our goal is to find a sequence (i) whose memory peak is smaller than the available memory on the GPU $M_{\text{GPU}}$, and (2) whose overall duration is minimal. In order to make the problem tractable, we introduce additional assumptions

1. Offloading before loss: all offloading operations must complete before the computation of the loss and prefetching operations (each prefetching operation being performed only once) start only after the computation of the loss. This is a reasonable assumption since offloading (resp. prefetching) introduce delays and therefore should be performed as early (resp. as late) as possible.

2. Memory persistency holds: if for some layer $i$ activation $\bar{a}^i$ is computed, then no operations related to layers $i'$ with $i' < i$ take place until $B_i$. This is a common assumption in the rematerialization literature, and it was shown in [3] that in all practical cases, optimal solutions satisfy this assumption (despite a theoretical result showing that there are cases where this is not true)

3. Asynchronous and continuous offloading: activations should be offloaded/prefetched entirely, but memory allocation and release can be performed in several steps during the communication. This assumption proved to be efficient in [4] to provide a tractable context without degrading the solution quality, and indeed without this assumption, the problem becomes strongly NP-hard. Continuous offloading thus allows to write the dynamic program to obtain efficient solutions; however these solutions are evaluated in Section 5 in a context where the tensors are discarded entirely and at once.

In Section 4, our goal is to find the optimal sequence of operations *under above assumptions*. Removing some of these assumptions may therefore improve the solution (by seeking the solution in a larger space), but a priori at the cost of more expensive computations.

1. A first limitation comes from the ability to offload $\bar{a}$ activation, which is not easy to implement in frameworks like PyTorch, because it requires modifying and accessing internal structures. Solutions that currently implement offloading, such as VDNN [32, 2], therefore require a thorough rewriting of the transfer operations.

2. A second limitation of our work is that it relies on the possibility of linearizing networks, *i.e.* representing them as a sequence of operations, in which each operation can in turn be arbitrarily complex (and associated to any type of dependences). In practice, most networks are easy to sequentialize. In the case of rematerialization only, optimal solutions based on integer linear programming such as Checkmate [18] were proposed. However, the cost

of integer linear programming limits their use to very small networks (smaller than those considered in Section 5) and their relaxation come without any guarantee. The extension of the proposed approach to complex dependency structures is nevertheless not straightforward.

# 4 Combination of offloading and rematerialization

We propose a dynamic programming to optimally solve the problem of finding a sequence containing rematerialization and offloading operations of minimal duration. This algorithm is denoted by `pofo`, for "Persistent with Offloading during Forward Only". To save space, we will not detail in the main paper all the equations of the dynamic program, which involves a large number of cases. We will focus in the main part of the paper on the intuitions and the general working principle of the dynamic program and refer the reader to Appendix B for detailed derivations and proofs.

## 4.1 Rationale of the different operations

The `pofo` algorithm is based on a sequence of choices, that consist in deciding, for each $1 \leq i \leq L$ (i) which type of operation $F_i$ we are going to use and whether we are going to compute $a^i$ or $\bar{a}^i$ (ii) whether we are going to keep the input value $a^{i-1}$ or $\bar{a}^{i-1}$ in the memory of the GPU, offload it in the memory of the CPU or completely delete it from the memory and (iii) how to compute $B_i$.

Concerning (i), the size of $\bar{a}^i$ is in general larger than $a^i$. Both can be used to compute $F_{i+1}$ and $B_{i+1}$, but if $a^i$ only is kept in memory (either CPU or GPU), then $F_i$ will have to be recomputed during the backward phase before $B_i$. Concerning (ii), if we delete $a^{i-1}$ or $\bar{a}^{i-1}$ from memory, then it will be necessary to recompute $F_{i-1}$ before processing $B_i$, but it will save memory for the whole subsequent sequence $F_{i+1} \dots F_L B_L \dots B_{i+1}$. If $a^{i-1}$ or $\bar{a}^{i-1}$ is offloaded on CPU memory (which may take some time), the memory will be available for the subsequent sequence, and $a^{i-1}$ or $\bar{a}^{i-1}$ will have to be prefetched before computing $B_i$. The interest of offloading obviously depends on the bandwidth $\beta$ of the PCI bus. Concerning (iii), if the input activation ($a^{i-1}$ or $\bar{a}^{i-1}$) has been deleted from memory, we will have to recompute it, starting from the last kept activation (either $a^k$ or $\bar{a}^k$ for some $k < i - 1$). Several rematerialization sequences are possible to do this operation that have potentially different durations and that offer different prefetching possibilities, depending on their memory profile, and the choice of the optimal sequence will depend on the memory state before computing $B_i$, as detailed in Appendix B.

## 4.2 Intuition of the overall scheme and state variables

The dynamic programming solution is built in several steps. We consider separately the forward propagation within which the offloading of activations can be done. The forward propagation is followed then with the loss calculation synchronized with the end of offloading operations. After loss computation, the backward propagation interleaved with the prefetching starts.

The general principle of `pofo` is the following. During the forward phase (before the computation of the loss), we consider the layers in an increasing order (from 1 to $L$). At each step, we have to decide which operation to implement among $F_{all}^\ell$, $F_{ck}^\ell$, $F_{\varnothing}^\ell$ and in the case where the input activation is kept, we have to decide if we add $O_\ell$. Inspired by the ideas of Automatic Differentiation [13] (but proposed in the context of a fully homogeneous chain without offloading), our dynamic programming relies on the following remark: any solution can be decomposed into parts, where each part computes layers $i$ to $j$ ($j \geq i$), and only the input to layer $i$ is stored in memory. Starting from a layer $i$, whose input needs to be saved in memory, `pofo` needs to decide whether the input will be saved with $F_{ck}^i$ or $F_{all}^i$. It also needs to decide the index $j$ of the end of the corresponding part in the solution, so that $j + 1$ starts the next part of the sequence, and its input will be the first activation kept in memory after $i$. A recursive call to the sub-chain starting from $j$ allows to obtain the corresponding running time. Between the two saved activations of layers $i$ and $j$, a sequence of $F_{\varnothing}^k$ takes place. As forwards $F_{ck}^i$ and $F_{all}^i$ correspond to two different behaviors, they generate two different cases of the dynamic programming that are considered in Appendix B.1. In addition, an offloading decision needs to be made for the input of layer $i$, either store it in the memory of the GPU or offload it to the CPU; this decision impacts the memory available for the rest of the chain and idle times due to communications.

In order to use dynamic programming, we need to define a set of variables that describes the state of the system at any instant. This set of variables should be as small as possible, since it has a direct

influence on the size of the data structure and on the computing time to solve the dynamic program. On the other hand, these variables must be chosen wisely and they must contain enough information to make decisions for subsequent layers and be updated according to these decisions. To evaluate correctly memory constraints and to compute idle times, it is important to know how memory may vary between layers $i$ and $j$ with saved inputs: from some minimal memory occupation till maximal memory occupation. Both can be described for the forward and the backward phases using only three state variables: in addition of the index $i$ of the last layer with saved input discussed above, we use $A_i, \Delta_{F_i}$ and $\Delta_{B_i}$, in addition to a boolean variable $x$.

- $A_i$ denotes the total GPU memory occupied by the saved values among $a^0, \ldots, a^{i-2}$ and $\bar{a}^1, \ldots, \bar{a}^{i-2}$ that are not transferred to the CPU.

- $\Delta_{F_i}$ denotes the amount of data from $a^0, \ldots, a^{i-2}$ and $\bar{a}^1, \ldots, \bar{a}^{i-2}$ that the schedule still needs to offload after $F_{i-1}$.

- $\Delta_{B_i}$ denotes the amount of data from $a^0, a^1, \ldots, a^{i-2}$ and $\bar{a}^1, \ldots, \bar{a}^{i-2}$ that the schedule should prefetch before starting $B_{i-1}$.

- $x$ is a boolean specifying whether the input of layer $i$ is $a^{i-1}$ (if $x = 0$) or $\bar{a}^{i-1}$ (if $x = 1$).

We denote as $\mathcal{I}_i^x = (1 - x)a^{i-1} + x\bar{a}^{i-1}$ the size of the input activation for layer $i$. With these state variables, we can compute $M_{F_i}^x$, the memory on the GPU after executing $F_{i-1}$, and $M_{B_i}^x$, the memory on the GPU before executing $B_{i-1}$ (excluding $\delta^{i-1}$) are given by $M_{F_i}^x = A_i + \Delta_{F_i} + \mathcal{I}_i^x$ and $M_{B_i}^x = A_i + \Delta_{B_i} + \mathcal{I}_i^x$.

These memory values represent the maximal memory occupation between layers $i$ and $j$ for the forward and backward phases respectively, and are used for computing idle times when overlapping communications with computations, according to Lemma 1. On the other hand, to estimate the feasibility of the scheduled operations, we also need to know the memory after everything has been offloaded, which is either $A_i + \mathcal{I}_i^x$ (when $\mathcal{I}_i^x$ stays on the GPU) or $A_i$ (when $\mathcal{I}_i^x$ is sent to the CPU), which we use to obtain the maximal memory available on the GPU.

The updates of the state variables follow simple rules described below

- if no new data is offloaded (prefetched), then $\Delta_{F_i}$ ($\Delta_{B_i}$) is constantly decreasing from index $i$ to index $j$ at speed $\beta$, until reaching zero;

- if new data has to be offloaded (prefetched), then the data size is added into $\Delta_{F_i}$ ($\Delta_{B_i}$);

- $A_i$ is updated if new data is saved on the GPU (without being later offloaded to the CPU).

Our goal is to find the sequence of operations that minimizes the overall execution time. However, both rematerialization and offloading can induce extra time with respect to the execution of the sequence $F_1 \ldots F_L B_L \ldots B_1$ which can be computed given infinite memory on the GPU:

- **During the forward phase**, offloading helps to keep GPU memory low, but, if the transfers to the CPU are not fast enough due to limited bandwidth, some idle time might occur, waiting for enough memory to be freed by offloading. This is analyzed in details in Appendix B.1.

- **At the interface between the forward and the backward phase**, we need to enforce that offloading and prefetching are well synchronized by the computation of the loss, what can in turn introduce some idle time on the GPU. This will be analyzed in Appendix B.2.

- **During the backward phase** (after the computation of the loss), there might be two sources of delays: recomputations of discarded activations and idle times induced by prefetching. Indeed, some prefetching operations may not be completed by the time the activation is needed, if they are delayed because of memory constraints. In fact, these two sources of extra times are not independent, since a longer rematerialization sequence with lower memory needs might allow more overlapping of prefetching with computations and thus avoiding idle time later. We propose an auxiliary dynamic program that includes the computation of intermediate idle times in each recursive call to find the best schedule under prefetching. The idle times for prefetching are found using Lemma 1. This is analyzed in Appendix B.3.

Finally, we present a lemma that describes the general scheme for optimally overlapping the communications with scheduled operations. Lemma 1 is valid for both prefetching and offloading, as their behavior is symmetrical. Offloading takes place at the beginning of the sequence, making available memory increase at the speed $\beta$ from its initial value $m_{min}$. On the contrary, prefetching is done at the end of the sequence, making available memory decrease at the speed $\beta$ until reaching $m_{min}$. Performing offloading as soon as possible and prefetching as late as possible allows to have more available memory for the middle of the execution. Note that $m_{min}$ could be equivalently replaced with $M_{\text{GPU}} - M_{\text{max}}$, where $M_{\text{max}}$ denotes the maximal memory occupation (either $M_{F_i}^x$ or $M_{B_i}^x$).

**Lemma 1** *Let us consider a fixed sequence of operations, for which the available memory increases (resp. decreases) during execution because of data unloading (resp. prefetching), with its minimum at $m_{min}$. Let us denote by $\mathcal{M}_o^{\mathcal{S}}$ the memory required to process operation $o \in \mathcal{S}$, and $d_o$ the distance between $o$ and the end (beginning) of sequence $\mathcal{S}$,i.e. the cumulative duration of operations taking place before (after) operation $o$. Then, the execution of $\mathcal{S}$ needs to be delayed by some idle time*

$$\epsilon = \max\left(\frac{\max_{o \in \mathcal{S}}(\mathcal{M}_o^{\mathcal{S}} - \beta d_o) - m_{min}}{\beta}, 0\right).$$

The above mentioned ingredients can be used to build the dynamic programming algorithm `pofo`, solving optimally the problem in the pseudo-polynomial time. Theorem 1 states this result, while the complete description of the dynamic program and the proof of the theorem can be found in Appendix B. The algorithm is pseudo-polynomial in the memory limit and it relies on discretization of the memory values. In the experiments of Section 5, we systematically use 50 discretization steps (scaling so that $M_{\text{GPU}} = 50$), which shows no discretization effect (we tried more steps to confirm this) and a reasonable complexity, with an execution time below 4 minutes in the worst cases. Since optimization is performed only once before the training starts, we argue that this is fully acceptable.

**Theorem 1** *Under assumptions of Section 3.2, the problem of finding the minimal processing time for the chain from Figure 1, using operations from Table 1 together with offloading $O_\ell$ and prefetching $P_\ell$, under memory limit $M_{GPU}$ discretized with $N_{GPU}$ values and bandwidth $\beta$, can be solved optimally with a dynamic programming algorithm with a complexity of $O(L^2 N_{GPU}^3 + L^3 N_{GPU}^2)$.*

## 5 Experiments

### 5.1 Additional heuristics

We have implemented two other sophisticated heuristics for comparison purposes. These heuristics are not bound with the assumptions of Section 3.2, but come without any optimality for the produced sequences.

In the first heuristic, called `opportunist`, the objective is to use the communication medium as much as possible. We compute the first layer with $F_{all}$ and offload its input and output, ensuring that the memory will remain fully available for the rest of the computation. The next layers are computed with $F_\varnothing$ until the end of communications. We then start a new communication by performing the next layer with $F_{all}$ and offloading its input and output, and so on until the end of the sequence can be entirely performed in memory. This process thus builds *groups* of layers between two $F_{all}$ operations. We then compute the backward phase for each group using the `rematerialization` algorithm, and concatenate them (with the necessary prefetches) to obtain the final sequence. Note that `opportunist` is conceptually close to the implementation from Superneurons [36].

The second heuristic is called `autocapper`. It relies on an internal `capper` algorithm, that uses an increased memory limit $M' > M_{\text{GPU}}$ as an additional input. `capper` computes a pure rematerialization sequence with the limit $M'$, finds the peak memory usage in the sequence and offloads the lowest indexed activation present in GPU memory at that time. This process is repeated until the sequence fits in the memory limit $M_{\text{GPU}}$. `autocapper` calls `capper` with 40 values of $M'$, evenly spaced between the target $M_{\text{GPU}}$ and $M^{\text{high}}$, the memory required without rematerialization or offloading. For each value $M'$, the resulting sequence from `capper` is simulated and the best one is kept. Note that, when `autocapper` does not perform recomputations, it behaves as the GREEDY heuristic in [4].

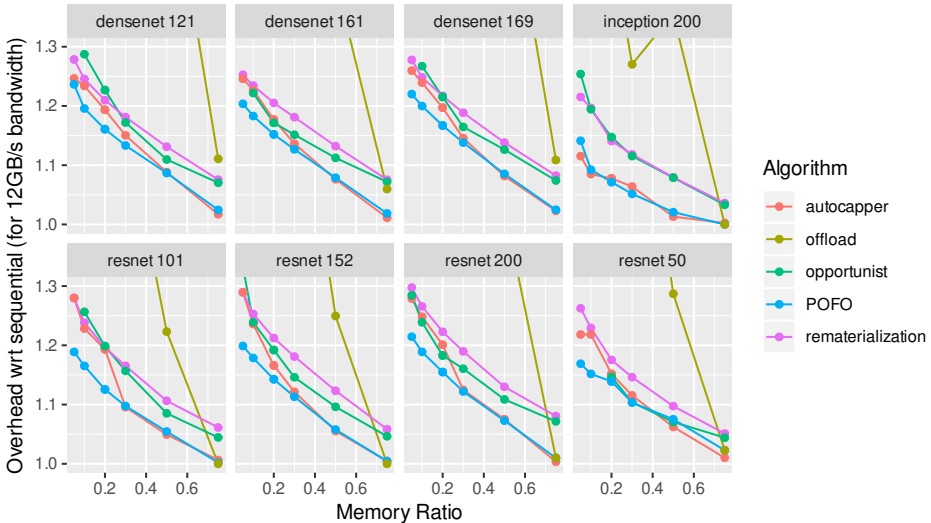

Figure 2: Simulation results for fixed bandwidth $\beta = 12$GB/s and varying memory ratio $\alpha$.

## 5.2 Simulation results

We measured running times and memory occupation of several networks from PyTorch `torchvision` package: `resnet`, `densenet` and `inception`, with a batch size of 16 and images of $500 \times 500$ pixels. For simplicity, we only present a single value for batch size: we also experimented with other batch and image sizes, and obtained very similar results. Time measurements were performed on a NVidia Tesla V100 GPU. We also measured the bandwidth obtained when transferring PyTorch tensors from and to the GPU and obtained 12GB/s. The simulation results presented here were obtained using 4 cores of a 24-core Haswell Intel® Xeon® E5-2680 v3 at 2,5 GHz, with 128GB of memory, and used about one hour of computation. We modified the open-source rematerialization framework `rotor` to handle offloading, and we implemented our algorithms in this framework.

We consider 5 algorithms in total: our dynamic program `pofo`, the optimal `rematerialization`-only algorithm from [3], the optimal `offload`-only approach (DYNPROG heuristic from [4]), and `autocapper` and `opportunist` heuristics. We use the `sequential` time as a reference: it is the time that it would take to process forward and backward phases with infinite memory, *i.e.* the sum of all forward and backward times. For each network, we compute the highest and lowest memory requirements (denoted respectively $M^{\text{high}}$ and $M^{\text{low}}$): $M^{\text{high}}$ is obtained with the `sequential` approach, while $M^{\text{low}}$ is obtained by recomputing everything from the beginning at each step of the backward phase. We can thus explore the whole range of achievable memory sizes for this network, by considering values within the interval $[M^{\text{low}}, M^{\text{high}}]$: for a given ratio $\alpha \in [0, 1]$, the memory limit $M_{\text{GPU}}$ is set to $(1 - \alpha)M^{\text{low}} + \alpha M^{\text{high}}$. We consider values of $\alpha$ from 0.05 (low memory scenarios) to 0.8 (high memory scenarios). Larger values of $\alpha$ are not included since this makes the memory limit so large that the optimization problem becomes easy, and all algorithms behave very similarly, with almost zero overhead. Depending on the networks, the values of $M^{\text{low}}$ range from 1 to 2.5GB, and the values of $M^{\text{high}}$ range from 4 to 20GB. On the other hand, the sizes of the weights range from 50 to 250MB, which highlights that the main memory usage for these networks comes from the activations. For the algorithms which perform offloading, we also vary the bandwidth value $\beta$, from 12GB/s (a realistic scenario) to 36GB/s (corresponding to possible future improvements).

The results are shown on two plots: Figure 2 presents the behavior of the algorithms for fixed bandwidth and varying memory constraint, while Figure 3 explores the effects of increasing the bandwidth for a given memory limit. We draw the following conclusions:

- When memory is large ($\alpha = 0.75$), offloading is more effective than rematerialization: with $\beta = 12$GB/s, there is time to offload and prefetch enough data to run the whole chain. However, for smaller memory limits, the `offload`-only policy exhibits much worse performance than `rematerialization` (unless more bandwidth is available, see Figure 3).

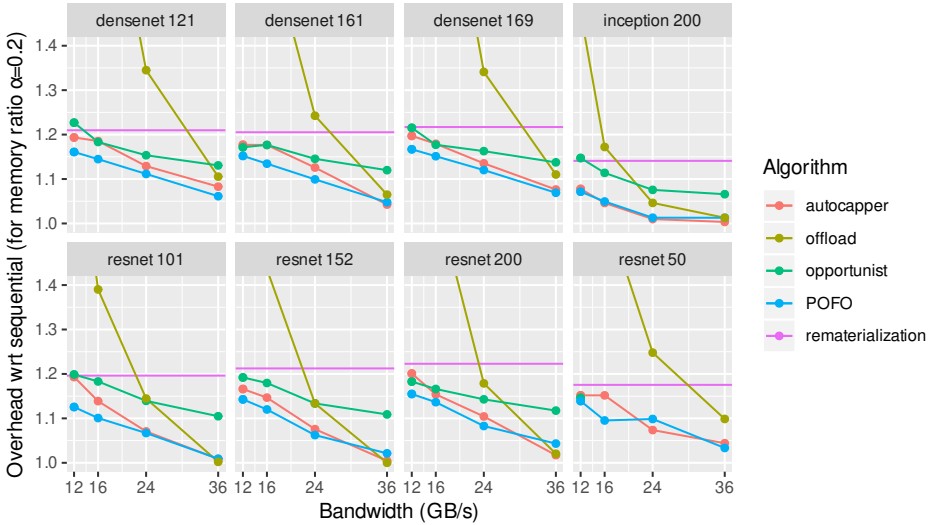

Figure 3: Simulation results for fixed memory ratio $\alpha = 0.2$ and varying bandwidth.

- Unless in some cases with small memory, the `opportunist` policy achieves slightly better performance than pure `rematerialization`, but is sometimes unable to produce a solution when the memory limit is too low. On the other hand, the more sophisticated `autocapper` algorithm obtains very good performance for memory ratios above $0.5$.

- The `pofo` algorithm successfully combines the advantages of both rematerialization and offloading, and consistently outperforms both of them. When the memory limit is high enough, the optimization problem is relatively easy, and `autocapper` and `pofo` achieve similar performance. In some cases `autocapper` is marginally better than `pofo`, as `pofo` produces solutions under assumptions of Section 3.2, while partial memory releases (assumption (iii)) are difficult to implement in practice. For lower memory limits, the more optimized `pofo` algorithm is able to produce much better sequences.

- As can be seen on Figure 3, the `offload`-only approach works perfectly when the bandwidth is high. Indeed, if there is enough communication capability to offload all the data while it is produced, it is possible to avoid recomputations without inducing idle time. Compared to pure offloading techniques, our `pofo` solution allows to train models with large activations with cheaper communication links.

A more precise analysis shows that the solutions computed by `pofo` offload a significant amount of data, about 25% of the total activation size for the `inception` and `resnet` networks, and 20% for `densenet`. This represents for example 2.1GB of offloaded data for `resnet 101`. For the same example, the amount of discarded data (which is recomputed later through rematerialization) varies from 0 (for $\alpha = 0.75$) to 1.3GB (for $\alpha = 0.1$). In general, the reason why `pofo` is more efficient than offloading alone is that rematerialization reduces memory usage, which means that fewer activations need to be offloaded. This allows to fit the offloading and prefetching time within the computing time. Symmetrically, the reason why `pofo` is more efficient than rematerialization alone is because offloading allows to remove some activations from the GPU memory "for free" in terms of recomputation. The rematerialization procedure thus needs to recompute fewer layers, which reduces the overhead cost.

## 5.3 Experimental results

In the recent release of PyTorch 1.10, the introduction of the `saved_tensors_hooks()` feature makes it possible to implement the offloading technique described in this paper. Indeed, this function allows to register hooks that can capture all tensors generated by an operation and then "pack" them (compress or offload) during the forward propagation and "unpack" them (extract or prefetch) during the backward propagation. We have implemented a preliminary version of our best performing

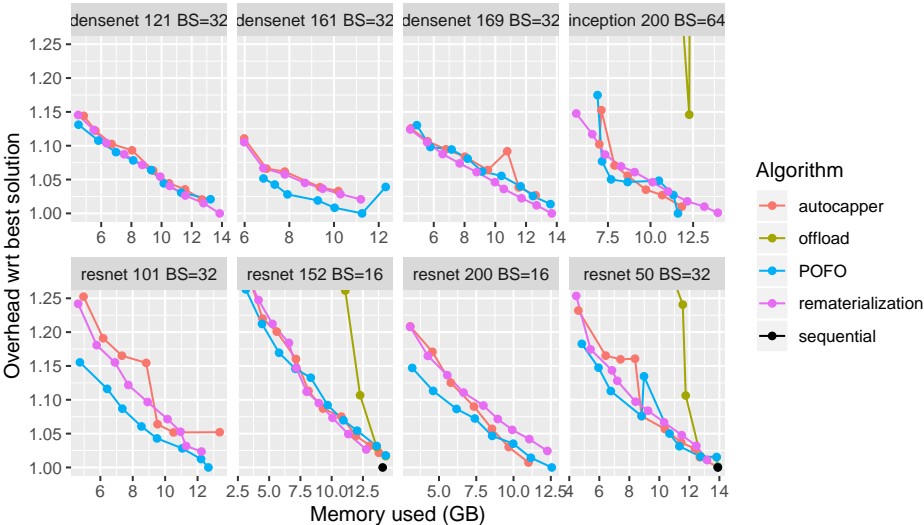

Figure 4: Experimental results on actual runs.

algorithms (`pofo` and `autocapper`) and made them available in `rotor` [1]. We further present the current results in Figure 4 obtained by testing these new extensions of `rotor` on the same neural networks as in the previous section, this time on an actual V100 GPU.

Overall, these preliminary experiments confirm that combining offloading with rematerialization allows to significantly improve over pure rematerialization in most scenarios. In particular, `pofo` is still the best algorithm for memory optimization among the ones considered in the plot. In most cases, `pofo` either shows the smallest overhead or behaves in the same way as other methods. However, `densenet-169` is the exception to this trend, where `rematerialization`-only is strictly better than `pofo` or `autocapper`, but even in this case `pofo` outperforms `autocapper`. On the other hand, the case of `resnet-101` demonstrates the significant improvement of `pofo` over other strategies, highlighting that there exist cases for which the benefit is important.

This discrepancy in the results obtained in simulation and in the experiments comes from the fact that the current implementation of `pofo` and `autocapper` in `rotor` uses more memory than expected. Our simulation results hint that further optimization of this preliminary implementation should allow to obtain even better performance.

## 6 Conclusion

In this paper, we formalize the problem of the optimal combination of rematerialization and offloading, which are two classical strategies for coping with memory limitations on a GPU. We show that the optimal solution can be computed using dynamic programming, in a few seconds or a few minutes for very deep networks. From experiments, we show that the combination of offloading and rematerialization is very efficient and allows, in many cases, to transparently perform training with 4 to 6 times less memory, at the cost of a 10-20% time overhead. Among the perspectives of this work, one could be interested in minimizing the energy consumption under both memory and throughput constraints. This study would enable to quantify the relative effects of data transfers and redundant computations from a different viewpoint. Another perspective of this work would be to adapt the dynamic programming approach to the case where we consider the possibility of offloading not only the activations, but also the weights of the network, which is necessary in the case of recent very large NLP networks in particular and which is used, for example, on the basis of heuristic optimization in Deepspeed [3]. The dynamic programming approach also seems to be adapted to this new context. Finally, it could be possible to extend the results of this paper to more general memory hierarchies, which include the case of CPU with limited memory and a possibility of offloading to a disk.

---

[3] `https://www.deepspeed.ai`

## Acknowledgments and Disclosure of Funding

We would like to thank our reviewers for their useful feedback and insights helping to improve our work. We would like to express special thanks for pointing out the new useful function of PyTorch that made our implementation possible. This work was supported by The HPC-BigData Inria Project Lab (IPL).

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
