# OpenReview forum: "Efficient Combination of Rematerialization and Offloading for Training DNNs"
_NeurIPS.cc/2021/Conference — NeurIPS 2021 Poster_

### Official Review · Reviewer_39eR · 2021-07-06

**Rating:** 6
**Confidence:** 4

**Summary:**

The paper presents an algorithm, called pofo, for optimizing the execution time of DNN training on a single GPU, considering a combination of rematerialization and offloading strategies.

**Limitations And Societal Impact:**

Yes

**Main Review:**

Major Comments
1.	Rematerialization and offloading have been already proposed in the literature. It is not clear which are the challenges of combining both approaches, if any. Therefore, the contributions do not seem much novel.
2.	In practice, can the proposed framework adapt to a limited memory size as a given constraint?
3.	Does the proposed method work only with feed-forward networks, or also with recurrent networks?
4.	In Section 5.2, “In some cases autocapper is marginally better than pofo, as pofo produces solutions under assumptions of Section 3.2, while partial memory releases (assumption (iii)) are difficult to implement in practice.” Why are there different assumptions for the autocapper procedure? For comparison purposes, the assumptions should be the same for every algorithm. Moreover, the reasons why the autocapper outperforms the proposed pofo should be discussed more comprehensively.

Minor Comments
1.	There is no need to report Figure 1 and Table 1 in the appendix, since they are already reported in the main manuscript.


**Time Spent Reviewing:**

4

---

> ### Author Response · Authors · 2021-08-10
> **Answer to Reviewer 39eR**
>
> **About limited memory size (we assume this means limited CPU memory):** In this case, a natural solution would be to use an additional level in the memory hierarchy, using for example the possibility to offload on NVMe. In the context of rematerialization only and for a completely homogeneous chain (corresponding to the classical usage case in Automatic Differentiation), such an extension to hierarchical memory has been considered in https://doi.org/10.1145/3378672. In the homogeneous case, the problem is nevertheless simpler since there are closed formulas, whereas the problem is NP-complete in the heterogeneous case which corresponds to the case of DNNs. It seems a priori possible to extend the dynamic programming approach that we propose to this context, by adding variables to describe the state of the system as in the multi GPU case, but here too, the dynamic programming will become too computationally expensive, unless we succeed in showing structural properties or in finding reasonable assumptions to simplify it.
>
> **About recurrent networks:** The use of rematerialization for recurrent networks has been considered for example in https://papers.nips.cc/paper/6221-brains-on-beats.pdf. The problem is different: on the one hand, it is easier because the networks are homogeneous, as in the context of automatic differentiation, for which closed forms have been proposed for the rematerialization problem. On the other hand, it is more difficult because one has to compute the solution for different network length, since the number of layers depend on the input sequence.
> pofo can in fact be directly used in the case of recurrent networks. Indeed, one can directly use pofo to compute the optimal solution for different sequence lengths (the cost will be significantly more important than for a feed forward network but it is paid only once before the training). Nevertheless, it is likely possible to use the homogeneity property to reduce the computational cost. For example, it might be possible to combine the closed form formulas derived by the automatic differentiation community to offloading, but this combination is not trivial, and we leave this approach for future works.
>
> **About different assumptions for autocapper:** We want to emphasize that the comparison made in the experimental section has the same setting for all algorithms. The theoretical analysis of the POFO algorithm requires additional assumptions to be able to prove an optimality result. However, when we simulate the resulting solution, we use the same assumptions as for all other algorithms.
> The autocapper algorithm is a simpler heuristic, whose solutions sometimes do not satisfy the assumptions of POFO. In some situations, this allows autocapper to provide a solution which is better than the one produced by POFO.
>
> **About the novelty of our contributions:** Rematerialization and offloading are two NP-complete problems in the weak sense, so it is indeed natural to use dynamic programming to solve them.
>
> A first main difficulty in this setting is to find a relevant and minimal set of variables to keep the complexity reasonable. For example, the fact (l245-248) that one can determine $M_{F_i}$ and $M_{B_i}$ from the variables $A_i$, $\Delta_{F_i}$ and $\Delta_{B_i}$ is critical in practice to save a variable and make the computations possible.
> Another difficulty is to find a set of assumptions that allow to simplify the problem while keeping good performance: for example, the linearization of the networks and the assumption on asynchronous and continuous offloading allow to control the complexity of the optimization problems. The first assumption of Section 3.2 (offloading before loss) is actually very helpful in reducing the size of the search space, and the experiments show that it still allows to obtain good solutions.
>
> The previous question on the use of a two-stage memory hierarchy is a good illustration of this difficulty. It might be "easy" to derive a dynamic programming in this extended setting, but without fresh ideas for choosing variables and proving additional structural properties, it is likely that the resulting dynamic programming will only be able to solve problems of very small size.
>
> Furthermore, relative to the previous Dynamic Program for offloading (reference [5] in our paper), we can highlight the following contributions:
> + We propose a general formula (Lemma 1) for overlapping any number of operations with data transfer (either offloading or prefetching);
> +Merging prefetching and backward phase is not trivial. In [5], the sequence to perform the backward operations is predefined: just the sequence of backward operations without forwards. In pure rematerialization, the sequence is generated assuming that the value of maximal available memory was constant. In the present paper, we need to generate the sequence while the maximal available memory may change over time. Thus, the resulting sequence may be different from the one from the classical rematerialization problem, as it tries to adapt to the changing memory limit due to prefetching. Moreover, we need to take the idle times from data transfers into account. It was possible only with the help of the special dynamic programming described in Appendix B.3. When designing it, limiting the number of variables was also an important concern, thus we needed to apply the additional structural property suggested in Lemma 1.
> + Together with the dynamic program pofo, we propose a smart heuristic autocapper, that proved to be more competitive than a straightforward approach opportunist.

---

### Official Review · Reviewer_nNiv · 2021-07-16

**Rating:** 7
**Confidence:** 4

**Summary:**

This paper tackles the problem of combining rematerialization with offloading by using dynamic programming, and demonstrate (in simulations) that their approach can substantially outperform pure offloading as well as pure rematerialization in terms of runtime in different situations.

**Limitations And Societal Impact:**

Yes

**Main Review:**

Overall, I like the problem that this paper tackles - how do you balance various strategies for making your model bigger/faster? All of these different approaches have different tradeoffs, and it's often unclear how you should use one vs. the other. So, papers that attempt to resolve these questions are quite valuable.

The DP itself is fairly straightforward but reasonable.

My primary qualm with this paper is that the experiments are all performed with a simulator. In my opinion, the DP in this paper is not novel enough to have intrinsic novelty. However, the DP algorithm can still be a valuable contribution if they demonstrate that the DP actually works in practice. Much of the contribution of DP algorithms in situations like this is demonstrating that the assumptions made by the algorithm are realistic enough for the DP to result in a good solution. However, if you're simply running it on a simulator, then I'm not sure this successfully tests whether your algorithm's assumptions are realistic.

Regardless, I think that the simulator results are quite interesting (with good baselines), and demonstrate that offloading may be a technique that people trying to reduce activation memory should consider looking into.

To be honest, I am quite split on this paper. On the one hand, I think that this paper tackles a valuable problem, demonstrates some novelty in combining rematerialization + offloading, and has intriguing simulator results. On the other hand, without experiments on real hardware or actual software, I am skeptical that these paper's results will have much impact in the community.

**Time Spent Reviewing:**

4

---

> ### Author Response · Authors · 2021-08-10
> **Answer to Reviewer nNiv**
>
> **About the realism of simulation-only results:** We first want to emphasize that our simulator does not make the same assumptions as the ones used in the dynamic program (Section 3.2) and relies on a more realistic model. For example, in the simulations, offloading of an activation is performed in a single atomic step, which is more realistic and easier to implement in practice. Conversely, to make the offloading problem tractable (it is otherwise NP-Complete in the strong sense) we assume that offloading in several blocks is possible when we compute the solution using dynamic programming.
>
> The main reason for using a simulator for this work is that we have reused rotor software for rematerialization, which is based on Pytorch; and it is very difficult to offload $\bar{a}$ activations in Pytorch (other offloading solutions either use TensorFlow or their own framework directly in CUDA; DeepSpeed has some support for offloading activations, but only $a$ activations). We have tested (on real hardware) a version of POFO where it is only possible to offload $a$ activations, but this does not provide significant gain wrt pure rematerialization.
> This version (actual implementation, but of $a$ activations only) also allowed us to check that the results provided by the simulator are actually very close to the behavior on real hardware, which makes us confident that our simulation results for the $\bar{a}$ activations are relevant.
>
> **About the novelty of our contribution:**
> Rematerialization and offloading are two NP-complete problems in the weak sense, so it is indeed natural to use dynamic programming to solve them.
>
> A first main difficulty in this setting is to find a relevant and minimal set of variables to keep the complexity reasonable. For example, the fact (l245-248) that one can determine $M_{F_i}$ and $M_{B_i}$ from the variables $A_i$, $\Delta_{F_i}$ and $\Delta_{B_i}$ is critical in practice to save a variable and make the computations possible.
> Another difficulty is to find a set of assumptions that allow to simplify the problem while keeping good performance: for example, the linearization of the networks and the assumption on asynchronous and continuous offloading allow to control the complexity of the optimization problems. The first assumption of Section 3.2 (offloading before loss) is actually very helpful in reducing the size of the search space, and the experiments show that it still allows to obtain good solutions.
>
> The questions raised by Reviewer ZiAj on the multi-GPU case and Reviewer 39eR on complex memory hierarchies are good illustrations of this difficulty. It might be "easy" to derive a dynamic programming in such extended settings, but without fresh ideas for choosing variables and proving additional structural properties, it is likely that the resulting dynamic programming will only be able to solve problems of very small size.
>
> Furthermore, relative to the previous Dynamic Program for offloading (reference [5] in our paper), we can highlight the following contributions:
> + We propose a general formula (Lemma 1) for overlapping any number of operations with data transfer (either offloading or prefetching);
> +Merging prefetching and backward phase is not trivial. In [5], the sequence to perform the backward operations is predefined: just the sequence of backward operations without forwards. In pure rematerialization, the sequence is generated assuming that the value of maximal available memory was constant. In the present paper, we need to generate the sequence while the maximal available memory may change over time. Thus, the resulting sequence may be different from the one from the classical rematerialization problem, as it tries to adapt to the changing memory limit due to prefetching. Moreover, we need to take the idle times from data transfers into account. It was possible only with the help of the special dynamic programming described in Appendix B.3. When designing it, limiting the number of variables was also an important concern, thus we needed to apply the additional structural property suggested in Lemma 1.
> + Together with the dynamic program pofo, we propose a smart heuristic autocapper, that proved to be more competitive than a straightforward approach opportunist.

---

> > ### Comment · Reviewer_nNiv · 2021-08-18
> > **Thanks for the comments**
> >
> > I've read the other reviewer's comments as well as the responses.
> >
> > > About the realism of simulation-only results
> >
> > I understand that the assumptions of the simulator are different than the one used in the DP. However, I remain worried that the potential gap between the simulator and the "real world" will cause potential issues. Especially since, if I'm understanding this correctly, the original Rotor paper was not done using a simulator?
> >
> > I also don't really understand why we wouldn't see significant performance gains if we restrict ourselves to looking at a activations. Perhaps I'm not understanding the difference between a and a_bar - could you provide an example of that?
> >
> > I'm also a bit confused why offloading is so difficult to implement in PyTorch - it seems that this PR (https://github.com/pytorch/pytorch/pull/61928) implemented it as an autograd hook, which seems possible to do for the authors. It's possible that this is related to not understanding the difference between a and a_bar.
> >
> > Overall, I don't find the concerns of the other reviewers to be as important as the question of: "Will this systems paper, which proposes a dynamic programming solution, be a significant contribution if it's not providing code that can run their algorithm on actual devices?". If the simulator used was a widely adopted simulator with benchmarks demonstrating its fidelity, I would be less concerned.
> >
> > I'd be interested in hearing the other reviewers' thoughts, since it appears that other reviewers did not bring up this concern in their initial review.

---

> > > ### Author Response · Authors · 2021-08-23
> > > **Experiments and interest of Pytorch PR on offloading**
> > >
> > > **Interest of Pytorch PR on offloading**
> > >
> > > Thank you very much for your feedback! We had checked the PR about offloading at the time of submission but not recently. The one you point to (https://github.com/pytorch/pytorch/pull/61928) is exactly what we need. It was not available at the time we wrote the paper. This PR requires as we anticipated to dive deep into the C++ code of Pytorch.
> > >
> > > There is now a priori nothing to prevent us from offloading the $\bar{a}$ and we have started to modify Rotor to have a fully functional version of the code using ```torch.autograd.graph.save_on_cpu()```. Unless there are unforeseen difficulties, we should be able to do it in the next few days: as we mentioned it, the code architecture to offload the ${a}$ values is already written, we just need to enrich it.
> > >
> > > Concerning the simulation results, we have already compared the simulation code for checkpoint + offload of ${a}$ values and the real experiments and there is no significant difference. If the overhead of the ```torch.autograd.graph.save_on_cpu()``` code to offload $\bar{a}$ activations is not too big (according to the PR, it shouldn't be), we don't expect to see much differences, but of course we need to do the experiments to check it.
> > >
> > > Thanks again for pointing out this PR, we will come back to you with first results as soon as possible.
> > >
> > >
> > >
> > > **Remark on why only offloading $a$ activations does not provide significant performance gains:**
> > >
> > > To perform a Backward step, the $\bar{a}$ activation is required. Obtaining an $\bar{a}$ activation from an $a$ activation requires to perform a $F_{all}$  step, but $\bar{a}$ is necessary to compute the Backward step. Therefore, if we offload an $a$ activation, then we will have to run again the forward task that produced it in order to do the backward… Thus, the interest with respect to rematerialization is tiny. It is not null since the $a_l$ activation can be used to recompute $\bar{a_{l+1}}$ and Backward(l+1) before recomputing $\bar{a}_{l}$, so we indeed observe an improvement when offloading $a$ value, but it is a small one. When offloading $\bar{a}_l$, we can immediately compute the backward on the prefetched activation, without any extra computation. In this case, the interest  of offloading  is much more significant. This is the basis of our equations and simulation results, and thanks to ```torch.autograd.graph.save_on_cpu()```, it should also be possible in experiments.

---

> > > > ### Author Response · Authors · 2021-08-27
> > > > **First Experimental Results**
> > > >
> > > > Thanks again for pointing out this PR on offloading! Thanks to it, as we had anticipated, we were able to have a working version of POFO which allowed us to obtain the first experimental results, depicted on https://docdro.id/Y7AqSmV and that can be seen as the experimental counterpart of Figure 2 in the paper.
> > > >
> > > > Due to time constraints, we could only experiment POFO on a single experimental platform, for which the PCIe bus bandwidth is close to 12GB/s (but lower for small messages). These are therefore not the best conditions to assess the efficiency of the combination of rematerialization and offloading, since we can see in Figure 3 of the paper that a bandwidth of 16 or 24 GB is better beneficial to the combination of offloading+rematerialization (w.r.t. rematerialization alone).
> > > >
> > > > Nevertheless, the experimental results obtained with POFO (and the AutoCapper heuristic) are qualitatively very comparable with those in Figure 2. Moreover, in its current state, the code is functional but it is not yet optimized and there are probably performance gains to be expected, especially in terms of memory usage.
> > > >
> > > > Of course, if you want to play with it, we can anonymously make the code available to you as a tar.gz !

---

> > > > > ### Comment · Reviewer_nNiv · 2021-08-28
> > > > > **Thanks for the updated experiments!**
> > > > >
> > > > > Thanks for the rapid turnaround! I appreciate the effort that's been put into the experiments, and it addresses my most significant concern with the paper.
> > > > >
> > > > > Of course, I'd like to see more extensive experiments across a wider range of experimental platforms, and as with Reviewer ZiAJ, I'd like to see results on language models if possible (since those are some of the largest users of gradient checkpointing). But those can wait for the camera ready :)
> > > > >
> > > > > However, since this addresses my primary concern (simulator fidelity), I'll raise my score to a seven.
> > > > >
> > > > > By the way, to clarify my point about the PyTorch PR, it was *not* required to implement it within C++. If you look at the PR itself, it's done entirely in Python and as an autograd hook (https://github.com/pytorch/pytorch/pull/61928/files). My main point in bringing up the PR was to push back against the author's claims that it was infeasible to implement the strategy in PyTorch without C++ changes.
> > > > >
> > > > > Of course, it's unreasonable to expect all authors to be aware of all framework extensibility points. But I'm glad that the authors were able to use the PR to provide a real implementation during the rebuttal period.

---

### Official Review · Reviewer_ZiAJ · 2021-07-18

**Rating:** 7
**Confidence:** 5

**Summary:**

The paper tackles the memory bottlenecks of by activations that limits scaling of model and batch sizes in DL training. The approach is to discover the optimal combination of activation re-materialization and offloading to CPU memory that will minimize the runtime overhead. The optimal solution is found using a proposed dynamic programming algorithm, called `pofo`, which determines an optimal sequence of re-materialization and offload decisions for the network layers, under the constraint of GPU memory and communication bandwidth to CPU. The paper also presents simulation-based results showing that `pofo` more efficiently reduces activation memory pressure compared to different SOTA approaches for different image models and different memory and communication bandwidth constraints.

**Ethical Concerns:**

No.

**Limitations And Societal Impact:**

Yes.

**Main Review:**

This paper tackles a very important problem because GPU memory is perhaps the biggest bottleneck in DL since it grows at significantly slower rates compared to model, data, and compute.  Also, addressing activation memory, which is one of the major sources of memory consumption, is very impactful to the community. A solution that consists of optimally combining SOTA techniques is very appealing and the automated nature could realistically save the massive wastage of human and machine resources on tuning SOTA techniques.

The paper is very well written and I found it enjoyable to read end-to-end. I felt the authors did such a great job in how they used available space fit enough high-level intuition and low-level details such that I could fully grasp the paper without referring to the appendix.

Concerns:
1. A key weakness is that evaluation does not consider multi-gpu scenarios, which are more common for large-model training than single gpu. This seems like an unnecessary omission that is not adequately explained. Moreover, since your results are simulation-based it does not seem like much extra work is required. It seems that all that is needed is to consider PCI bandwidths per GPU lower than 12GB/sec (e.g., 3GB/sec per GPU for 16 GPUs in a DGX-2 box). This way you won't need to adjust your algorithm to consider inter-gpu communication (which is typically fast within a node anyways). Please clarify if I am missing anything.
2. I can't figure out if your results apply to language models. The key difference of language models that could affect the effectiveness of your approach is that model and optimizer consume much more memory leaving very little room for activations. On the other hand, and perhaps as a consequence, the batch size is typically much smaller than image models. So perhaps these two factors will cancel out and your results will still hold, but it is hard to tell since you don't have language model results. Another view is that perhaps your work is more desperately needed in the language model space where batch sizes are heavily constrained by memory pressure. Can you share any thoughts on this? Do the claims in the paper need adjusting to reflect the image-centric nature of the results?

Suggestions/questions:
1. The evaluation should include model and activation sizes, as well as the GPU memory size for the presented results.
2. Offloading to NVMe is already possible using ZeRO-Infinity and Nvidia GPUDirect (either from CPU memory or directly from GPU memory). This would be needed when CPU memory is not large enough for the activations. Can your approach can be extended for this?

**Time Spent Reviewing:**

4

---

> ### Author Response · Authors · 2021-08-10
> **Answer to Reviewer ZiAj**
>
> **About multi-gpu training:** The multi-GPU problem is indeed very interesting.
> A first solution, as suggested by the reviewer, consists in sharing *a priori* the bandwidth in G equal parts between the GPUs. In this case, pofo can be directly used to compute the optimal sequence (the same for all GPUs). We propose to include multi-GPU results using this approach in the final version of the paper.
> A second more general and more difficult solution consists in not partitioning a priori the bandwidth.  Indeed, in this case, it can be more beneficial to compute different solutions for each GPU, in order to avoid that the competition for the bandwidth induces idle time. The direct use of dynamic programming seems impractical, because the state variables would have to be duplicated  (to represent the state of all GPUs) and the induced computation time would be much too long. Thus, it would first  be necessary to prove structural properties to limit the search space and make the computation time tractable. We propose to keep this second multi-GPU version for future works.
>
> **About language models and heavy weights scenarios:** If the cost of storing the weights is very high, as it is the case in recent NLP models, this induces a high memory pressure that may prevent even a mini-batch of size 1 from being processed. In this case, the approach proposed in the paper and relying on offloading and rematerialization are particularly useful, because it can enable to train a model which otherwise would not have been possible.
>
> In the ZeRO paper  doi:10.1109/SC41405.2020.00024, it was considered how to train large neural networks such as Megatron GPT (very large language model) and despite the fact that model weights require significant memory space, storing activations is not negligible either (it represents 60GB for GPT-2 model). ZeRO uses another approach to reduce the memory consumption by distributing the load among all available resources (training is done on a multi-node cluster), which relies on a high bandwidth between the nodes. Our approach is orthogonal to that and could help to reduce memory even more efficiently.
>
> As a future works of this paper, we will propose in the conclusion (see the answer to the remark of the reviewer 1) to consider offloading of the network weights in addition to the activations, but the problem is a bit different and our dynamic programming approach cannot be directly extended to this context.
>
>
> **About offloading to NVMe:** The case of a bounded CPU memory size is also very interesting.
> For example, in the context of rematerialization only and for a completely homogeneous chain (corresponding to the classical usage case in Automatic Differentiation), such an extension to a memory hierarchy has been considered in https://doi.org/10.1145/3378672. However, in the homogeneous case, the problem is significantly simpler (there are closed-form formulas for the basic rematerialization problem), whereas the heterogeneous version (which corresponds to DNNS) is NP-complete. It seems *a priori* possible to extend the dynamic programming approach that we propose to a memory hierarchy context, by adding variables to describe the state of the system (as in the multi-GPU case). However, once again, the dynamic programming algorithm would become too computationally expensive, unless we succeed in showing structural properties or in finding reasonable assumptions to simplify it.

---

### Official Review · Reviewer_Zjqt · 2021-07-18

**Rating:** 6
**Confidence:** 4

**Summary:**

This paper investigates the advantages of _combining_ the memory optimizations of rematerialization and offloading for training large models on single GPUs.  In doing so it studies when such a combination is useful and unsurprisingly shows that a combination of the two provides better benefits than either optimization used in isolation while incurring a modest 20% runtime overhead.

**Limitations And Societal Impact:**

Limitations are mostly covered in my review above, but succinctly: the paper could do a better job explaining how offloading is used in conjunction with rematerialization - that is not just the effect of using them, but what is the sequence of offload/rematerialization operations are the algorithms in the paper suggesting (some sort of temporal view for intuition or percentage of tensors offloaded versus rematerialized for quantitative data).

**Main Review:**

I enjoyed reading your paper and your focus on memory saving optimizations for activations to push the boundaries of single GPU training.  I was however quite confused by all the discussion on offloading weights (lines 109-115, references to deepspeed and such).  My suggestion: just explicitly state up-front that your focus is on activations (and you assume weights and optimizer state fit in memory) and just ignore all this distracting discussion on offloading of weights.

I was a little surprised that when it came time to evaluate you picked a batch size of 16.  Could you provide a rationale as to why you pick this batch size for all the DNNs you evaluate with?  Is this something that you can fit for _sequential_ for each of the 8 DNNs you evaluate with? I assume you do this so that you can measure overhead with respect to sequential.  And if so, when you throw your memory saving optimizations at it, are you leaving memory on the GPU unused?  It is ok if you are doing this just for evaluation, but it would be good to call this out.

Is there a reason why your graph x-axis (alpha) ends at 0.8?  Why not show us what happens at 0.9 and 0.95?

One thing that seemed like a missed opportunity in studying this combination of optimizations was reporting what percentage of activations were offloaded versus what percentage were recomputed.  I suspect that because offload is limited by PCIe throughput, offload is used sparingly and spread out by large temporal intervals so that prefetching these tensors can be hidden.  It would be great if you can provide some quantitative data and intuition for this.

As I am writing this review, I realized that I am assuming you are prefetching tensors that you had offloaded (and that you are not doing this fetch in the critical path).  When you offload, are you assuming that you have space buffers available to prefetch data into?  That is, from the total memory available to you are reserving some memory for prefetch buffers?  And is offloading an option only because you are using rematerialization in between - that is you are increasing compute time and hence providing more of an opportunity to hide prefetching from slower CPU memory over PCIe (12 GB/s)?

Overall, while you show us that the combination of the two memory optimizations you consider are effective (compared to individual optimizations), it is unclear _why_ offloading is helping in this combination and how much offloading is being performed.  You have an opportunity to nail this analysis (even if it means suggesting offloading is used sparingly but should be used nonetheless in this combination).


**Time Spent Reviewing:**

4 hours

---

> ### Author Response · Authors · 2021-08-10
> **Answer to reviewer Zjqt**
>
> **About the discussion about weight offloading:** As suggested, we propose to delete the last paragraph of Section 2 (l109-115) and add a few lines of perspectives on offloading weights in the conclusion.
>
> **About the use of a single batch size of 16:** As mentioned in the paper (line 323): *We also experimented with other batch and image sizes, and obtained very similar results.* We chose to display only a single batch size in order to simplify the presentation, and to limit the computational cost of the experiments. We will add results for other settings (batch and image sizes) in the associated extended research report. It is indeed correct that in our evaluation, we fix the memory limit *a priori*, which may or may not correspond to the available memory on the GPU.
>
> **About the values of $\alpha$:** Initial experiments have shown that when $\alpha > 0.8$, the memory limit is so large that the optimization problem becomes quite easy, and all algorithms behave very similarly, with almost zero overhead. We have thus decided not to run complete simulations for these cases, in the interest of limiting the computational cost of the experiments. We can provide more results in the final version of the paper.
>
> **About the relative importance of offloading and rematerialization:** Offloading actually accounts for a significant part of the memory reduction. With 12GB/s bandwidth, the POFO algorithm offloads about 25% of the total activation size for the inception and resnet networks, and 20% for densenet. This represents for example 2.1GB of offloaded data for Resnet 101. For the same example, the amount of discarded data varies from 0 (for large memory limit, alpha=0.75) to 1.3GB (for alpha=0.1). If enough space is available, we can include additional plots and discussion with this information in the final version of the paper.
>
> **About buffers:** In our model, memory is reserved when the prefetch starts, not earlier. All algorithms make sure that the prefetch operations are performed only after enough memory has been freed by other compute operations. We do not use special prefetch buffers: data is prefetched directly in the correct place in memory.
>
> **About *why* offloading is helping, and about offloading only being an option because rematerialization increases compute time:** Actually, this is not exactly the case: we can see on Figure 2 that even with 1/5 of the required  memory (without rematerialization), the extra computational time is generally around 20%, and such an increase is not large enough to significantly increase the interest of offloading alone. Furthermore, Figure 2 also shows that for $\alpha = 0.75$, pure offloading outperforms pure rematerialization, which shows that offloading can be an option even without increasing compute time through rematerialization (this is true both for large memory limits and for high bandwidth).
>
> The reason why offloading + rematerialization works better than offloading alone is mostly that rematerialization reduces memory usage, which means that fewer activations need to be offloaded. This allows to fit the offloading+prefetching time within the computing time.
>
> And the reason why offloading + rematerialization works better than rematerialization alone is because offloading allows to remove some activations from the GPU memory "for free" in terms of recomputation. The rematerialization procedure thus needs to recompute fewer layers, which reduces the overhead cost.
>
> We will include this discussion in the final version of the paper.

---

> > ### Comment · Reviewer_Zjqt · 2021-09-16
> > **acknowledgement of having read your response**
> >
> > Thank you for your response.  I wanted to leave a note here acknowledging that we have read and considered your response to our queries.  I do believe that your paper will benefit from explaining _why_ and _when_ certain optimizations work better - it will only help readers appreciate your techniques.  In the longer term (not for this rebuttal), I hope there is some way you can visualize which layers use rematerialization and which ones use offloading. Although orthogonal, I suspect many systems for large models (say e.g., Harmony, Hot OS 2021) can benefit from your insights!

---

### Author Response · Authors · 2021-08-10
**General comment for reviewers**

We thank the reviewers for their thorough reading and relevant comments. You will find below our detailed responses to the remarks, associated with each individual review.
We have not identified any significant misunderstanding in the reviews and we propose to take many suggestions in the final version of the paper and to propose new experiments to illustrate more completely the interest of the combination of rematerialization and offloading. Other very good extensions suggested in the reviews, in particular on parallelism and the use of a more complex memory hierarchy, require more elaborated algorithmic developments. We keep them in mind for future works and will present and discuss them in the final version, but we explain below why these issues are difficult and cannot be completely addressed in the final version.

---

### Decision · Program_Chairs · 2021-09-27

**Decision:**

Accept (Poster)

**Comment:**

This paper proposes an optimized algorithm to compute a sequence of forward / backward / offload / prefetch operations on activations that optimizes training throughput of linearized DNNs under memory constraints. The rebuttal solves the reviewers' concerns, and the reviewers unanimously agrees to accept the paper.